# Early Stage Breast Cancer: Does Histologic Subtype (Ductal vs. Lobular) Impact 5 Year Overall Survival?

**DOI:** 10.3390/cancers16081509

**Published:** 2024-04-15

**Authors:** Fatima Mubarak, Gabrielle Kowkabany, Reed Popp, Shivam Bansal, Syeda Hoorulain Ahmed, Seema Sharan, Kulkaew B. Sukniam, Swathi R. Raikot, Paola Berrios Jimenez, Kyle Popp, Harsheen K. Manaise, Emmanuel Gabriel

**Affiliations:** 1The Aga Khan University Medical College, Stadium Road, P.O. Box 3500, Karachi 74800, Sindh, Pakistan; 2Department of Chemical and Biological Engineering, The University of Alabama, Tuscaloosa, AL 35487, USA; gakowkabany@crimson.ua.edu; 3University of Florida College of Medicine, 1600 SW Archer Rd, Gainesville, FL 32610, USA; reedpopp@ufl.edu; 4Government Medical College & Hospital, Block C, 1157-B, Chandi Path, 32B, Sector 32, Chandigarh 160047, India; shivambansal.sb8@gmail.com (S.B.); seemasharan2198@gmail.com (S.S.); harsheen98@gmail.com (H.K.M.); 5Division of Plastic and Reconstructive Surgery, Department of Surgery, University of Florida, Gainesville, FL 32608, USA; hoorulain_ahmed97@hotmail.com; 6Duke University Medical Center, 10 Duke Medicine Cir, Durham, NC 27710, USA; kulkaew.sukniam@duke.edu; 7Emory University, 201 Dowman Dr, Atlanta, GA 30322, USA; swathi.raikot@emory.edu; 8University of Puerto Rico, 6, 2526, 601 Av. Universidad, San Juan, PR 00925, USA; paola.berrios2@upr.edu; 9Florida State University, 600 W College Ave, Tallahassee, FL 32306, USA; kylelpopp@gmail.com; 10Department of General Surgery, Division of Surgical Oncology, Mayo Clinic Florida, 4500 San Pablo Road, Jacksonville, FL 32224, USA; gabriel.emmanuel@mayo.edu

**Keywords:** breast cancer, histology, overall survival

## Abstract

**Simple Summary:**

The two major histological subtypes of breast cancer are ductal and lobular cancers. Histology impacts the biological behavior of breast cancer, so it is imperative to study its influence on the overall survival of breast cancer patients. The existing literature reports conflicting conclusions regarding the impact that histology might have on survival, so we aimed to study a large sample of patient data obtained from the National Cancer Database to determine if histology and several other clinical factors have an impact on the survival of early-stage breast cancer patients who have T1–T2 tumors, are clinically node-negative and have no metastatic activity. These results may help clinicians better understand the behaviors of different subtypes of breast cancer and will help them take a more tailored approach to breast cancer treatment.

**Abstract:**

Histology is an important predictor of the behavior of breast cancer. We aim to study the impact of histology on the overall survival (OS) of breast cancer patients. We studied 11,085 breast cancer patients diagnosed with T1–T2 tumors, clinically node-negative and non-metastatic, from 2004 to 2019 included in the National Cancer Database. Kaplan–Meier curves, log-rank tests and Cox regression models were used to study the impact of histology and other variables on OS. In our patient population, 8678 (78.28%) had ductal cancer (IDC), while 2407 (21.71%) had lobular cancer (ILC). ILC patients were significantly more likely to be older, Caucasian, have a lower grade at diagnosis and be hormone receptor-positive compared to IDC patients. There was no statistically significant difference in the 5-year OS of early stage ductal (16.8%) and lobular cancer patients (16.7%) (*p* = 0.200). Patients of Hispanic and African American origin had worse OS rates compared to non-Hispanic and Caucasian patients, respectively. For node-positive disease, HER2+ tumors and triple-negative tumors, chemotherapy had a positive influence on OS (HR 0.85, 95% CI 0.77–0.93, *p* = 0.0012). Histology did not have a significant impact on the 5-year OS of early stage breast cancer patients.

## 1. Introduction

Female breast cancer is the most prevalent cancer type worldwide. In 2020, 2.3 million women were diagnosed with breast cancer, and 685,000 deaths were reported globally [1]. Despite the high incidence of the disease, death rates have declined by 43% from 1989 to 2020. This decline can be attributed to the early detection and increased awareness of breast cancer and the availability of better treatment modalities [2]. According to the Surveillance, Epidemiology, and End Results (SEER) database, 3,886,830 women were living with breast cancer in the United States in 2020. In the 2013–2019 period, the overall relative survival rate for breast cancer patients was 90.8% with declining 5-year survival rates for higher stages of disease at diagnosis [3]. 

More than 95% of breast cancers are adenocarcinomas, and they all arise from the terminal duct lobular unit. The two most prevalent histologic types of invasive breast cancer are ductal carcinoma and lobular carcinoma. Invasive ductal carcinoma (IDC) arises from lactiferous ducts and constitutes around 55% of all breast cancers [4]. Invasive lobular carcinoma (ILC) arises from the milk-producing lobules, and it is the second most common histological subtype of breast cancer and accounts for 5–15% of all breast cancers diagnosed [5].

According to studies based on the SEER database, the incidence of invasive lobular carcinoma has significantly increased from 9.6% to 15.6% in 12 years (1987–1999), particularly in postmenopausal women, compared to the incidence rates of IDC, which have remained relatively stable throughout the years [6]. Compared to ductal cancer, invasive lobular cancer is more likely to affect older postmenopausal women, be bigger, be hormone receptor-positive and have a low to absent Her-2 expression [7]. Ductal and lobular cancers also have different patterns of metastasis, with ductal cancer usually spreading to the lungs, pleura and central nervous system (CNS) and lobular cancer involving the peritoneum, ovaries and gastrointestinal system [8,9,10,11]. 

The overall survival for patients diagnosed with lobular and ductal cancer has been previously studied, but the reported prognosis has been variable [12]. This study aimed to look at 11,085 patients with early-stage T1–T2 tumors of clinically node-negative, non-metastatic breast cancer diagnosed between 2004 and 2019 included in the National Cancer Database. We aimed to identify key clinical characteristics of both major subtypes of breast cancer and to determine if there is a significant influence of tumor histology on the overall survival of early-stage breast cancer patients.

## 2. Materials and Methods

Data for 11,085 patients diagnosed with early-stage breast cancer (T1–T2 tumors, clinically node-negative) between 2004 and 2019 were obtained from the National Cancer Database. The inclusion criteria were female sex, clinical T1 or T2 tumors, clinically negative lymph nodes and no metastasis. While patients who were male, clinically T3–T4, had inflammatory breast cancer, were clinically node-positive or had metastasis were excluded.

The data were summarized based on histology (ductal and lobular). Independent variables included age, race, ethnicity, treatment facility, grade, clinical T stage, lymphovascular invasion, estrogen receptor status, progesterone receptor status, Her2 receptor status, use of surgery, chemotherapy and radiation. Categorical variables were expressed as frequency and relative frequency, while continuous variables were expressed as the mean, median, and interquartile range (IQR). All associations were compared using Mann–Whitney U and Fisher’s Exact tests, respectively. 

Kaplan–Meier curves were used to summarize overall survival (OS) based on histology. Log-rank tests were used to determine if there was a significant difference between OS in patients diagnosed with ILC versus patients diagnosed with IDC, and 5-year OS rates and 95% confidence levels were recorded.

Cox regression models were generated to study the influence of variables, including histology, age, race, Hispanic ancestry, treatment facility, grade, clinical T-stage and treatment received by patients, on the overall survival of breast cancer patients. Hazard ratios, 95% confidence intervals and *p*-values for each predictor were recorded for analysis.

All analyses were conducted in RStudio v4.0.2 at a significance level of ≤0.05.

## 3. Results

### 3.1. Patient and Tumor Characteristics

From the 11,085 patients who were diagnosed with T1–T2, N0, M0 breast cancer from 2004 to 2019 included in the National Cancer Database, 8678 (78.28%) had ductal cancer, while 2407 (21.71%) had lobular cancer.

Table 1 describes the key differences in the clinical and biological characteristics of ductal and lobular tumors. Lobular carcinoma (1116/2407, 46.36%) was more likely to be diagnosed in the elderly population (age > 65 years) compared to ductal carcinoma (3653/8678, 42.09%) (*p* = 0.001). There was a statistically significant difference in the racial distribution of both histological subtypes (*p* = 0.004). The lobular group had a higher percentage of white patients (2102/2407, 88.80%) compared to the ductal group (7337/8678, 85.93%), while the black population was more commonly diagnosed with ductal (926/8678, 10.85%) compared to lobular cancer (211/2407, 8.91%). 

Patients diagnosed with lobular cancer were significantly less likely to present with grade III disease (12.77%) compared to patients diagnosed with ductal cancer (30.66%) (*p* < 0.001). Patients were also less likely to have lymphovascular invasion with lobular cancer (585/2407, 27.96%) compared to ductal cancer (2930/8678, 38.86%) (*p* < 0.001). Lobular tumors (2311/2407, 96.41%) were more likely to be estrogen receptor (ER)-positive than ductal tumors (7564/8678, 87.57%) (*p* = 0.001). 

Patients who were diagnosed with lobular cancer were significantly more likely to receive aggressive treatment compared to patients who were diagnosed with ductal cancer. They had higher rates of mastectomy compared to the ductal group (*p* < 0.001) and were also more likely to receive adjuvant chemotherapy (*p* < 0.001).

### 3.2. Overall Survival

A Kaplan–Meier curve was drawn to summarize the difference in the overall survival of early-stage breast cancer patients (T1–T2, N0, M0) diagnosed with ductal cancer versus patients diagnosed with lobular cancer (Figure 1 and Table 2). The median survival time for ductal cancer patients was 38.05 months (95% CI: 37.59–38.74), while the median survival time for lobular cancer patients was 37.39 months (95% CI: 36.11–38.47). There was no statistically significant difference in the median survival time of early-stage breast cancer patients based on histology in our population. In our cohort of early-stage breast cancer patients, there was no statistically significant difference in the 5-year overall survival of invasive ductal cancer patients (16.8%) and lobular cancer patients (16.7%) (*p* = 0.200).

### 3.3. Cox Regression Analysis

In our study population of early-stage breast cancer patients, lobular cancer patients had a slightly reduced survival time compared to patients diagnosed with ductal cancer, but this difference was not statistically significant (HR 1.026, 95% CI: 0.96–1.08, *p* = 0.404, Table 3). Increasing age also did not have any significant impact on the overall survival of breast cancer patients. 

Race had a significant effect on the survival of breast cancer patients. Black women were significantly more likely to have a shorter survival time compared to white women (HR 1.106, 95% CI: 1.01–1.20, *p* = 0.019). Women of Hispanic origin were also more likely to live for a shorter time after being diagnosed with breast cancer (HR 1.175, 95% CI: 1.05–1.31, *p* = 0.004) compared to women from non-Hispanic origins. Patients who were treated at comprehensive community and academic centers were significantly more likely to survive for a longer time compared to patients treated at community centers. 

The types of surgical treatment and radiation therapy received by the patient did not impact survival significantly; however, patients with node-positive disease, HER2+ tumors and triple-negative tumors received chemotherapy, and they had an improved survival time compared to patients who did not receive chemotherapy (HR 0.85, 95% CI: 0.77–0.93, *p* = 0.001).

## 4. Discussion

Breast cancer can be histologically classified into two major categories, including invasive ductal and invasive lobular carcinoma. With the increasing incidence of the lobular subtype in recent years due to increased use of hormone replacement therapy [13], it has become imperative to study the characteristics of this subtype compared to the more prevalent ductal subtype and to see if the histological type of tumor significantly impacts the overall survival of breast cancer patients. This information can help guide clinicians regarding detection and management strategies required for these specific subtypes of breast cancer. 

In our study, we aimed to compare the key patient and tumor biological characteristics of early stage T1–T2 tumors of clinically node-negative and non-metastatic ductal and lobular cancer to identify if significant differences in terms of presentation, tumor behavior and management exist between the two histology types. The early-stage comparison between ductal and lobular cancer among such a large cohort is a main strength of our study. In our population, women more than 65 years old were significantly more likely to develop lobular cancer compared to ductal cancer (*p* = 0.001). This finding can help clinicians have higher suspicions of lobular cancer in older women and could help them screen this population for signs of lobular cancer on their wellness visits and provide them with up-to-date education on this subtype of breast cancer. These steps ensuring early detection could help improve the survival rates of lobular cancer patients. We saw that early-stage lobular cancer was associated with a higher grade at presentation, had a lower risk of lymphovascular invasion and was more likely to be hormone receptor-positive compared to early-stage ductal carcinoma. All these associations reached clinical significance. We also concluded that women diagnosed with the lobular subtype of breast cancer were more likely to receive aggressive treatment like mastectomy instead of lumpectomy, and they were also significantly more likely to receive adjuvant chemotherapy compared to women diagnosed with ductal cancer (*p* < 0.001). 

Our findings are in line with the existing literature comparing the characteristics of ductal and lobular cancers. Oesterreich et al., looked at 33,622 patients included in the Great Lakes Breast Cancer ILC data consortium. In their patient population, 3617 patients were diagnosed with ILC, while 30,045 patients were diagnosed with IDC. They similarly discovered that lobular carcinoma was significantly more likely to develop in older women (age more than 61 years, *p* < 0.001). They also concluded that ILC was more likely to be estrogen receptor-positive compared to IDC (96% vs. 77%, respectively, *p* < 0.001). ILC was also significantly more likely to be progesterone-positive compared to IDC (81% vs. 67%, respectively, *p* < 0.001). In their patient population, women diagnosed with lobular cancer were also more likely to receive aggressive surgical treatment compared to women diagnosed with ductal cancer. In total, 60% of the women diagnosed with ILC received mastectomies compared to 50% of women with IDC (*p* < 0.001). However, in their patient cohort, women with ILC were less likely to receive chemotherapy compared to women with IDC (*p* = 0.014) [14]. The identification of these key characteristics of lobular carcinoma can help clinicians with the detection and management of this most common special subtype of breast cancer. 

The main aim of our retrospective review was to determine if the histology of the tumor has a significant impact on the overall survival of early-stage (T1–T2, N0, M0) breast cancer patients. In our study, the analysis revealed that there was no statistically significant difference in the median survival time of patients diagnosed with early-stage invasive ductal cancer (38.05 months, 95% CI: 37.59–38.74) and patients diagnosed with early-stage invasive lobular cancer (37.39 months, 95% CI: 36.11–38.47). We also did not find any significant difference in the 5-year overall survival of ductal cancer patients (16.8%) compared to that of lobular cancer patients (16.7%) (*p* = 0.200). These results are aligned with those of multiple previous studies that have found no significant impact of histology on breast cancer survival outcomes at short-term follow-up.

Contradictory reports regarding the impact of histology on survival of breast cancer patients have been previously published in the literature. There are multiple studies that have reported that the histology of breast cancer has a significant impact on survival [15,16,17], while other studies demonstrate that histology does not, in fact, have a strong impact on the survival of breast cancer patients. Dian et al., performed a multivariate survival analysis of 2058 eligible patients (non-metastatic and those who did not receive adjuvant chemotherapy and anti-hormonal treatment) from their database of 5689 female patients diagnosed with pure ILC or IDC, which suggested that tumor histology was an independent variable that significantly impacted the overall survival (*p* = 0.046). Their Kaplan–Meier survival analysis also revealed that patients diagnosed with ILC had a significantly improved overall survival compared to patients diagnosed with IDC (*p* = 0.030) [15].

Similarly, Wasif et al., looked at 263,408 women diagnosed with breast cancer without distant metastasis from 1993 to 2003, included in the SEER database. In their cohort, 27,639 (10.1%) women had ILC, while 235,769 (89.5%) had IDC. The 5-year disease-specific survival (DSS) was 90% for all patients diagnosed with ILC and 88% for all women diagnosed with IDC, and this finding was significant (*p* < 0.001). They also investigated the effect of histology on 5-year disease-specific survival based on stage, which also yielded a similar result. For T1, N0 disease, ILC patients had 98% 5-year DSS while IDC patients had 96% 5-year DSS (*p* < 0.001). For T2, N0 disease, ILC patients had 94% 5-year DSS, while IDC patients had 88% 5-year DSS (*p* < 0.001). This shows that the prognosis of ILC is significantly better than IDC even when the patients were stage-matched [16].

On the other hand, multiple large-scale studies have reported no significant difference in the survival outcomes of breast cancer patients based on histology even though the tumor and clinical characteristics of ductal and lobular cancers have many differences [18]. Korhonen et al., looked at 243 patients diagnosed with pure invasive lobular carcinoma, and 243 matched patients diagnosed with invasive ductal carcinoma. They discovered that the mean overall survival time did not differ significantly between the two groups. It was 12.54 years for ILC patients (95% CI 11.60–13.48) and 13.55 years for IDC patients (95% CI 12.58–14.53). They could not determine any significant difference in breast cancer-specific survival in both groups [19]. These results are interesting because even though lobular cancer is generally associated with good prognostic factors like hormone receptor positivity [20], there is no significant impact on overall survival compared to ductal cancer. 

Some recent studies have suggested that there may not be a significant difference in the survival of ILC patients compared to that of IDC patients in the short term. When patients are followed for a long term (more than 10 years), lobular cancer may show worse prognosis compared to ductal cancer [21,22]. These findings bring up the need for more large population studies that follow lobular cancer patients long-term and can determine if there is a significantly worse long-term survival outcome of ILC compared to IDC. This would help clinicians determine appropriate adjuvant treatments and follow-up surveillance for patients diagnosed with lobular breast cancer. 

We also generated Cox regression models to study the impact of different tumor and patient characteristics on the overall survival of early-stage breast cancer patients with no distant metastasis. Our analysis revealed that lobular histology had a slightly negative impact on overall survival compared to ductal histology in T1–T2, clinically node-negative patients, but this effect was not statistically significant (HR 1.026, 95% CI 0.96–1.08, *p* = 0.404). Even though older women were at a higher risk of developing lobular cancer, age at the time of diagnosis did not significantly impact overall survival.

The race of the patient proved to be a significant factor in the overall survival of breast cancer patients. African American women were more likely to have a shorter survival time compared to Caucasian females (HR 1.106, 95% CI 1.01–1.20, *p* = 0.019). This finding is in line with previously published literature regarding survival in black breast cancer patients [23,24,25]. This disparity is attributed to decreased access to quality healthcare and screening mammography that may be due to multiple social and economic reasons, including mistrust in the healthcare system, poor quality counseling provided to black women regarding breast health, a lack of health insurance, no childcare, etc. [26]. Breast cancer patients of Hispanic origin were also more likely to have a worse overall survival outcome compared to women of non-Hispanic origin [27,28,29]. This outcome can be attributed to the fact that Hispanic populations are more likely to belong to a lower socioeconomic background, have lesser access to good education and are less likely to have health insurance. Hispanic patients are also more likely to be at a higher stage at diagnosis and have hormone receptor-negative breast cancer, which leads to worse overall prognosis [26].

Race is an important factor considered in the survival outcomes of breast cancer patients. It is known that African American women with breast cancer have higher mortality rates and shorter survival times compared to Caucasian women. This may be due to disparities in access to healthcare and due to the increased prevalence of basal-type (ER/PR+, HER2−) breast cancer in black women [30]. According to the Carolina Breast Cancer Study (CBCS), survival outcomes for Luminal A-type breast cancer (ER/PR+, HER2−) were worse for African American women compared to Caucasian women, but interestingly, despite the higher prevalence of basal-type cancer in black women, the survival outcomes for basal-type were slightly worse in white women compared to black women. This suggested that triple-negative breast cancer may not be biologically more aggressive in black women compared to white women. Thus, factors like lower access to healthcare, lower socioeconomic status, a lack of insurance and higher stage of presentation at diagnosis may play an important role in the worse survival outcomes of black women [31].

Even though in our patient cohort there was no significant difference in patients who received treatment at community centers versus academic centers based on histology, we did find that breast cancer patients who were treated at academic centers were more likely to have an improved overall survival (HR 0.76, 95% CI 0.70–0.84, *p* < 0.001). This finding makes sense because academic centers are more likely to have ample resources for cancer treatment and would be more likely to use the latest research-based treatment regimens compared to community centers, which are usually limited in resources.

In our population of early-stage, non-metastatic breast cancer patients, the type of surgical management or radiation therapy did not play a significant role in the overall survival of breast cancer patients. The SINODAR-ONE clinical trial was conducted to study the role of axillary lymph node dissection (ALND) in patients undergoing breast conservation surgery or mastectomy who had T1–T2 tumors who were clinically node-negative and had 1–2 positive sentinel lymph nodes. Patients were randomized to an ALND arm or to a no-further-surgery arm. Tinterri et al., analyzed the patients in the ALND arm to determine the characteristics that predicted four or more involved axillary lymph nodes. One of the independent predictors of extensive axillary disease in early-stage breast cancer patients (T1–T2, N0, M0) was lobular histology compared to ductal histology (OR 4.185, 95% CI, 1.284–1.443, *p* = 0.041) [32].

In our patient population, patients with node-positive disease, HER2+ tumors and triple-negative tumors received chemotherapy, and they had an improved length of survival time compared to similar patients who did not receive adjuvant chemotherapy (HR 0.85, 95% CI 0.77–0.93, *p* = 0.001), which is in line with the current literature [33,34,35]. It is a limitation of our study that we were unable to study the effect of hormonal therapy on overall survival because it has been previously proven that in postmenopausal women, hormonal therapy alone is beneficial for lobular cancer because it is usually hormone receptor-positive, while hormonal therapy along with chemotherapy has survival benefit for patients with ductal cancer [36]. This further proves that histology is an important factor to consider while devising a treatment plan for patients diagnosed with early-stage breast cancer. 

Even though our study included a large population of early-stage, non-metastatic breast cancer patients in a national database, the retrospective nature of the study and missing data reduce the generalizability of our findings. Our study only included patients with T1 and T2 tumors who were clinically node-negative with no metastasis, so our results are not applicable to patients with higher stages of disease or patients with distant metastasis. Furthermore, we were unable to study the effect of mixed lobular and ductal histology on the overall survival of breast cancer patients, which reduces the broad applicability of our results. Another limitation of our study is that we only considered a 5-year follow-up for overall survival (as opposed to disease-specific survival). Since other studies have suggested that ILC may have worse survival outcomes compared to IDC in the long term (more than 10 years) [37,38], studies following breast cancer patients for longer periods with more endpoints of cancer-specific survival are warranted. We were also not able to study other predictors of survival like disease-free survival, metastasis-free survival, etc. 

It is important to conduct long-term follow-up studies regarding overall survival in lobular and ductal cancer patients to correctly determine the clinical behavior of these subtypes of cancer and to develop more accurate guidelines regarding treatment and the long-term surveillance of breast cancer patients.

## 5. Conclusions

While histology may impact the management and long-term follow-up of breast cancer patients, in our study, there was no difference in 5-year overall survival between patients with early-stage ductal or lobular invasive cancer. For early T1–T2 and N0 tumors, management can be similar, which was supported by our study’s findings. While there is a perception that lobular cancer can be more aggressive, this seemed to not be the case among our early-stage cohort. However, it is important to note limitations in the application of our findings to more advanced diseases (e.g., node-positive or metastatic disease), and it is important to note that our outcome was overall survival as opposed to recurrence-free or disease-specific survival. Studies with longer follow up times (more than ten years) and more in-depth analyses of survival outcomes are needed to further study the difference between lobular and ductal cancer and to help clinicians provide good quality, evidence-based treatment to breast cancer patients.

## Figures and Tables

**Figure 1 cancers-16-01509-f001:**
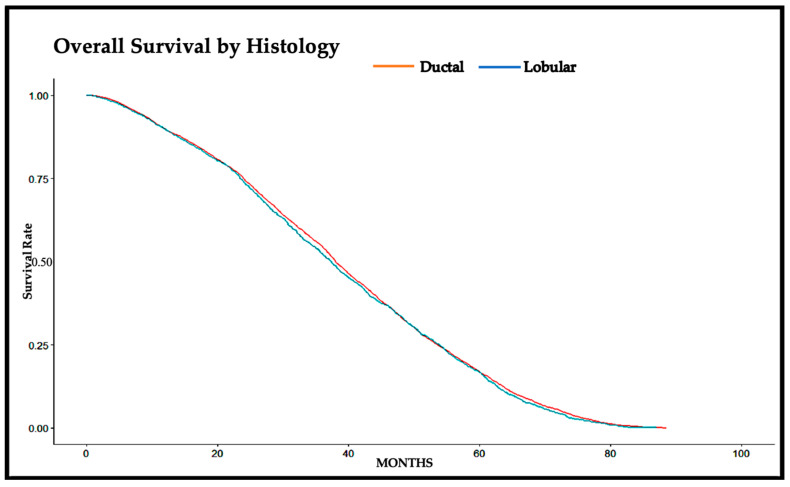
Overall survival of patients diagnosed with ductal versus lobular cancer.

**Table 1 cancers-16-01509-t001:** Patient and tumor biological characteristics based on histology.

		Ductal	Lobular	Overall	*p*-Value
		8678	2407	11,085	
Age	<55	2389 (27.53%)	595 (24.72%)	2984 (26.92%)	0.002
	55–65	2636 (30.38%)	696 (28.92%)	3332 (30.06%)	
	>65	3653 (42.09%)	1116 (46.36%)	4769 (43.02%)	
Race	White	7337 (85.93%)	2102 (88.80%)	9439 (86.56%)	0.005
	Black	926 (10.85%)	211 (8.914%)	1137 (10.43%)	
	Native American	31 (0.363%)	5 (0.211%)	36 (0.330%)	
	Asian	244 (2.858%)	49 (2.070%)	293 (2.687%)	
Hispanic	No	8035 (94.93%)	2233 (94.98%)	10,268 (94.94%)	0.96
	Yes	429 (5.069%)	118 (5.019%)	547 (5.058%)	
Facility	Community Center	946 (10.90%)	240 (9.971%)	1186 (10.70%)	0.28
	Comprehensive Community Center	4145 (47.76%)	1136 (47.20%)	5281 (47.64%)	
	Academic Center	2160 (24.89%)	640 (26.59%)	2800 (25.26%)	
	Other	1427 (16.44%)	391 (16.24%)	1818 (16.40%)	
Grade	1	1681 (20.15%)	543 (23.66%)	2224 (20.91%)	<0.001
	2	4104 (49.19%)	1459 (63.57%)	5563 (52.29%)	
	3	2558 (30.66%)	293 (12.77%)	2851 (26.80%)	
T Clinical	T1	5750 (66.26%)	1556 (64.64%)	7306 (65.91%)	0.15
	T2	2928 (33.74%)	851 (35.36%)	3779 (34.09%)	
Lymphovascular Invasion	No	4609 (61.14%)	1507 (72.04%)	6116 (63.50%)	<0.001
	Yes	2930 (38.86%)	585 (27.96%)	3515 (36.50%)	
ER Status	Negative	1074 (12.43%)	86 (3.588%)	1160 (10.51%)	<0.001
	Positive	7564 (87.57%)	2311 (96.41%)	9875 (89.49%)	
PR Status	Negative	1728 (20.03%)	315 (13.17%)	2043 (18.54%)	<0.001
	Positive	6901 (79.97%)	2077 (86.83%)	8978 (81.46%)	
Her2 Status	Negative	5700 (85.88%)	1563 (83.99%)	7263 (85.47%)	0.044
	Positive	937 (14.12%)	298 (16.01%)	1235 (14.53%)	
Nodes Positive	Mean Median/IQR	1.195 1.0 (1.0, 1.0)	1.25 1.0 (1.0, 1.0)	1.207 1.0 (1.0, 1.0)	<0.001
Surgery	Lumpectomy	5853 (67.47%)	1375 (57.13%)	7228 (65.22%)	<0.001
	Mastectomy	2803 (32.31%)	1029 (42.75%)	3832 (34.58%)	
	None	19 (0.219%)	3 (0.125%)	22 (0.199%)	
Radiation	Adjuvant	5807 (66.92%)	1582 (65.72%)	7389 (66.66%)	0.25
	Intraoperative	58 (0.668%)	10 (0.416%)	68 (0.613%)	
	Neoadjuvant	9 (0.104%)	5 (0.208%)	14 (0.126%)	
	Perioperative	10 (0.115%)	1 (0.042%)	11 (0.099%)	
	None	2692 (31.02%)	783 (32.53%)	3475 (31.35%)	
	Unknown	102 (1.175%)	26 (1.080%)	128 (1.155%)	
Chemotherapy	Adjuvant	6966 (80.27%)	2004 (83.26%)	8970 (80.92%)	<0.001
	Neoadjuvant	235 (2.708%)	52 (2.160%)	287 (2.589%)	
	Perioperative	194 (2.236%)	54 (2.243%)	248 (2.237%)	
	None	829 (9.553%)	218 (9.057%)	1047 (9.445%)	
	Unknown	454 (5.232%)	79 (3.282%)	533 (4.808%)	

**Table 2 cancers-16-01509-t002:** The 5 yr survival rates of patients with ductal and lobular breast cancers.

	Median (Months)	0.95 LCL	0.95 UCL	5 Yr Survival Rate	*p*-Value
Ductal	38.05	37.59	38.74	0.168	0.20
Lobular	37.39	36.11	38.47	0.167	

**Table 3 cancers-16-01509-t003:** Overall survival cox regression analysis.

Feature		Hazard Ratio	95% CI	*p*-Value
Histology	Ductal	1.0		
	Lobular	1.026	0.96–1.08	0.40
Age	<55	1.0		
	55–65	1.047	0.98–1.11	0.15
	>65	0.9805	0.91–1.05	0.61
Race	White	1.0		
	Black	1.106	1.01–1.20	0.019
	Native American	1.169	0.75–1.79	0.48
	Asian	1.413	1.23–1.62	<0.001
Hispanic	No	1.0		
	Yes	1.175	1.05–1.31	0.004
Facility	Community Center	1.0		
	Comprehensive Community Center	0.8357	0.77–0.90	<0.001
	Academic Center	0.7694	0.70–0.84	<0.001
	Other	0.7742	0.70–0.85	<0.001
Grade	1	1.0		
	2	1.006	0.94–1.06	0.86
	3	0.9014	0.83–0.97	0.006
T Clinical	T1	1.0		
	T2	1.03	0.97–1.08	0.28
Surgery	Lumpectomy	1.0		
	Mastectomy	0.9613	0.89–1.02	0.24
	None	1.013	0.52–1.96	0.97
Radiation	No	1.0		
	Yes	0.9801	0.91–1.05	0.57
Chemotherapy	No	1.0		
	Yes	0.8528	0.77–0.93	0.001

## Data Availability

All data are included in this manuscript.

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
