# Peer review of "Early Stage Breast Cancer: Does Histologic Subtype (Ductal vs. Lobular) Impact 5 Year Overall Survival?"

_cancers, 2024, doi:10.3390/cancers16081509_

Round 1
Reviewer 1 Report
Comments and Suggestions for Authors
In this manuscript, the authors summarized and discussed the impact of ductal and lobular histology on overall survival in early-stage breast cancer. On this basis, they further investigated the key differences in clinical and biological characteristics of ductal and lobular tumors. This work provides some insights and opinions into breast cancer patients’ treatment. The manuscript is well-organized and stated relatively clear . I would suggest accepting it after the following minor concerns and addressed.
1. The anthors mentioned several times in the discussion section that “The results of this study are in line with the existing literature”, then, what is this study is different from others?
2. The description of data characteristics is suggested to be more concise, clear and coherent in the Materials and Methods section.
3. There is a misspelling of a word (analyses) in Line 113.
4. In page 3, there is an extra symbol after subheading 3.1, Please check the manuscript carefully.
5. I suggest to take out the three-line form in figure 1 and rendering it as Table 2. Doing so allows the reader to understand the data more clearly, reducing the possibility of confusion and distraction.
6. The picture in your paper are a bit blurry in Figure 1. Please consider replacing them with clearer ones.
7. All P values in the manuscript should be capitalized in italics.
8. All Tables are not in the right format and should be adjusted according to the standards.
Comments on the Quality of English Language
Minor editing of English language required
Reviewer 2 Report
Comments and Suggestions for Authors
The manuscript presents a very interesting study performer on a big group of breast cancer patients – over 11000 patients. The authors showed that there is no difference between patients diagnosed with ductal or lobular carcinoma in overall survival. However, much more interesting analyses were connected with the response to treatment and socioeconomic factors modifying the Hazard Ratio. The access to modern treatment and the insurance coverage of the expenses connected with the diagnosis and therapy are very important factors which significantly modify the survival of breast cancer patients.
Author Response
Thank you so much for taking out the time to review our manuscript, “IMPACT OF DUCTAL AND LOBULAR HISTOLOGY ON OVERALL SURVIVAL OF EARLY-STAGE BREAST CANCER (T1-T2, N0, M0) PATIENTS.” We really appreciate your overall positive comments.
Reviewer 3 Report
Comments and Suggestions for Authors
I worry about a 5 year overall survival endpoint when assessing ILC outcomes as there is data that shows metastatic recurrences are more common after 5 years with ILC and when overall survival is compared at 10 years, ILC actually has worse prognosis compared with IDC (https://ascopubs.org/doi/10.1200/JCO.2007.14.9336, https://www.sciencedirect.com/science/article/pii/S0960977621004203 among others).
In Table 1- for "facility" rather than having the numbers associated with the different treatment settings, I would recommend naming the type of treatment setting in the table. It won't take up more room but will improve the flow of the reader.
With respect to the histologic grade for ILC vs IDC, there is a mismatch in the paragraph describing the cohort on page 2/conclusion page 7 and Table 1. The text reports more grade 3 tumors in ILC and the table says the opposite. I suspect the table is the correct association.
For the cox regression analysis, it would be nice to have some commentary here on how race interacts with tumor subtype, socioeconomic factors as the correlations with outcome, race, treatment at academic vs community center are likely highly influenced by psychosocial factors and biologic subtype of disease beyond histology (ie more TNBC in Blacks and very little TNBC represented among ILC patients)
Reviewer 4 Report
Comments and Suggestions for Authors
The study entitled "IMPACT OF DUCTAL AND LOBULAR HISTOLOGY ON OVERALL SURVIVAL OF EARLY-STAGE BREAST CANCER (T1-T2, N0, M0) PATIENTS" performed by Mubarak F et al. analyzes the impact of ductal and lobular histologies on the overall survival (OS) of early-stage breast cancer patients (T1-T2, N0, M0) using data from the National Cancer Database covering 11,085 patients between 2004 and 2019.
The research finds no statistically significant difference in median survival times between patients with invasive ductal carcinoma (IDC) and those with invasive lobular carcinoma (ILC), nor in 5-year OS rates between the two groups.
Despite ILC being more common in older women, more likely to present with aggressive disease, and more often hormone receptor-positive compared to IDC, these histological differences did not significantly affect OS.
The present study definitely has some potential because it analyzes a very large population of clinically node negative, M0, breast cancer patients, and it is generally well written; however, I have some major concerns that need to be addressed before potential publication:
- The Title can be misleading to readers of the journal; in fact; from the results of the present study, histology DOES NOT have impact on overall survival. I suggest the authors to modify the title;
- In the Introduction section, page 2, lines 57-65, is too general and not related to the present study. I suggest to eliminate those lines;
- - In the Introduction section, page 2, line 88, please add the following references (PMID: 37384206, PMID: 11989148, PMID: 14566024) for providing further context on the metastatic potential of ILC;
- Materials and Methods section is too short overall. Try to expand it (especially the first part) and divide it in at least two different paragraphs. Please consider a Study design paragraph. Please consider to add a Figure showing a Flow Chart of the selection of patients process. Separate Statistical analysis in a different paragraph;
- In the Results section, paragraph 3.1, also add numbers (n) other than percentages;
- In the Discussion section, please add the following reference (Tinterri, Corrado MD*,†; Canavese, Giuseppe MD*; Gentile, Damiano MD*,†. To Dissect or Not to Dissect? The Surgeon’s Perspective on the Prediction of Greater Than or Equal to 4 Axillary Lymph Node Metastasis in Early-Stage Breast Cancer: A Comparative Analysis of the Per-Protocol Population of the SINODAR-ONE Clinical Trial. Annals of Surgery 5(1):p e405, March 2024. | DOI: 10.1097/AS9.0000000000000405 ) which found ILC to be a major predictor on the presence of 4 or more metastatic lymph node in clinically node negative T1-2 breast cancer patients (same population of your study). This can HIGHLY impact modern adjuvant combination therapies and prognosis;
- You must modify Conclusions. "Histology plays a crucial role in the diagnosis, management, and long-term follow-up of breast cancer patients. It is important to study the biologically distinct behavior of invasive lobular carcinoma compared to ductal carcinoma so that they can be managed accordingly.". From the results of your study this is not true! How do you manage breast cancer differently according to histology??? Do you perform breast conserving surgery or mastectomy based on the histology ??? Do you change chemotherapy regimens based on the histology ??? Please clarify.
Comments on the Quality of English LanguageIt needs moderate revisions.
Round 2
Reviewer 4 Report
Comments and Suggestions for Authors
The manuscript can be accepted in the present form